# Using Internet Search Data to Forecast COVID-19 Trends: A Systematic Review

**Simin Ma** , **Yan Sun and Shihao Yang** *

H. Milton Stewart School of Industrial and Systems Engineering, Georgia Institute of Technology, Atlanta, GA 30332, USA
* Correspondence: shihao.yang@isye.gatech.edu

**Abstract:** Since the outbreak of the coronavirus disease pandemic (COVID-19) at the end of 2019, many scientific groups have been working towards solutions to forecast outbreaks. Accurate forecasts of future waves could mitigate the devastating effects of the virus. They would allow healthcare organizations and governments to alter public intervention, allocate healthcare resources accordingly, and raise public awareness. Many forecasting models have been introduced, harnessing different underlying mechanisms and data sources. This paper provides a systematic review of forecasting models that utilize internet search information. The success of these forecasting models provides a strong support for the big-data insight of public online search behavior as an alternative signal to the traditional surveillance system and mechanistic compartmental models.

**Keywords:** COVID-19; internet search data; Google trends; forecasting model; statistical models

## 1. Introduction

Coronavirus disease 2019 (COVID-19), a contagious disease caused by the severe acute respiratory syndrome coronavirus 2 (SARS-CoV-2), has quickly spread worldwide and caused more than 630 million reported cases and 6.6 million reported deaths [1]. Since the initial outbreak, many variants of COVID-19 have emerged (alpha, delta, omicron, etc.), leading to drastic surges in confirmed cases, hospital admissions, and deaths, which have severely threatened the healthcare systems and resources [2,3]. Given the continuous developments of COVID-19 variants and variant-led infections, understanding the pandemic progression and disease dynamics is more urgent than ever. At the same time, accurate forecasts of the future disease progressions would assist efficient healthcare and financial resource allocation, as well as timely implementation of intervention policies [4]. Accurate and robust predictions will also help prepare against upcoming new variants and subsequent waves. Meanwhile, accurate predictions heavily depend on the underlying forecasting techniques and modeling efforts that efficiently assess the situations of the past, and thereby enable better predictions about the future. Among the weekly forecast reports compiled by the U.S. Centers for Disease Control and Prevention (CDC) [5], contributed to by various research groups and individuals around the globe, the prevailing COVID-19 forecasting approaches are based on machine learning techniques [6–8] and mechanistic models [9–11]. Meanwhile, data collected from various platforms, serving as inputs to the forecasting techniques, are also crucial in forecasting.

In particular, internet-based big data could be a valuable complementary data source to monitor disease progression, and directly integrate with traditional surveillance approaches and disease modeling efforts, shown by numerous studies in the past decades [12–19]. For instance, Google Flu Trend (GFT) [15], a digital disease surveillance system operated by Google that utilizes selected Google search queries' volume to estimate the future influenza-like illnesses (ILI) activities for more than 25 countries, was one of the first few studies that demonstrated public search behaviors' potential in complementing traditional forecasting

analysis. More recently, many studies have improvised and improved upon GFT and provided more robust and accurate real-time influenza activity estimates around the world, including machine learning models [13,20,21], statistical methods [12,16,18,22], compartmental models [23,24], etc. Many studies have also exploited public search behaviors to track other infectious diseases, such as dengue [17], HIV/AIDS [25], etc.

In this study, we present a systematic review and analysis of the forecasting efforts in the literature that focuses on harnessing public search behaviors through internet search data. To the best knowledge of the authors, there is currently no systematic review article that focuses on methods utilizing online search information for COVID-19 forecasting. Therefore, this study aims to summarize the extended application of internet search data to enable future research studies to reciprocate and contribute to the field of COVID-19 forecasting and infectious disease tracking more efficiently. Accurate and robust forecasting performances shown in the considered studies here demonstrate the predictive power of internet search data, and its potential in serving as early warning signals for subsequent COVID-19 waves.

This review study is organized as follows. Section 2 briefly introduces various existing COVID-19 forecasting models, as well as other related literature considered in this review study. Section 2 also briefly explains the selection criteria for the research studies included in this paper. Section 3 illustrates the data sources of internet search data, and other data used alongside the online search data. Section 4 demonstrates various models that exploit internet search data in the recent publications to model the spread of COVID-19 in various regions of the globe, as well as the evaluation procedure for results comparison. Section 5 summarizes the forecasting results and evaluations conducted by the selected research studies. Finally, in Sections 6 and 7, a discussion and conclusion of the entire review study are presented.

## 2. Related Studies

### 2.1. COVID-19 Forecasting Models and Related Literature

As COVID-19 prediction has attracted increasing interest from researchers, many prediction approaches have been proposed from various perspectives. There are two general infectious disease modeling directions: the mechanistic approach and the data-driven approach. Mechanistic models mathematically formulate the disease dynamics by dividing the population of interest into different compartments, such as susceptible, infectious, and recovered/removed, and model each compartment with ordinary differential equations [26]. The majority of mechanistic models used are based on the susceptible–exposed–infectious–recovered–deceased model (SEIRD) [27] and its variants. For instance, Yang et al. [28] propose a modified SEIR model by introducing population move-in and move-out parameters. He et al. [29] further split the infectious population into infectious with and without interventions, which indicates different contagious rates and recovery rates under various interventions implemented. The mechanistic approach can be effective for longer horizon predictions and disease dynamic understandings but might be inadequate for detecting sudden changes and subsequent outbreaks (due to COVID-19 variants) [30].

The data-driven approach treats COVID-19 prediction as a time series prediction task, using available historical data and dynamic social behaviors. These models are typically built upon statistical frameworks [31–33] and recent advances in machine learning [34,35] and deep learning algorithms [35–37]. For example, Kumar et al. [31] modeled the evolution of the COVID-19 outbreak, and performed prediction using autoregressive integrated moving average (ARIMA) and prophet time series forecasting models. Papastefanopoulos et al. [34] compared six different time series approaches using two publicly available datasets, showing that machine learning methods can accurately estimate the infectious rates in the future, given the data of actual testing for a small portion of the population. Er et al. [37] proposed a transformer-based method, a deep data-driven machine learning model adapted from natural language processing applications, to generate

2-week-ahead COVID-19 death forecasts for all U.S. counties, utilizing COVID-19-related cases, deaths, and community mobility trends.

Although many research studies focus on the aforementioned categories of approaches, this paper mainly focuses on another category of COVID prediction, which utilizes external sources of data to assist in more accurate predictions.

The correlation between online search data (such as Google Trends [38], Baidu [39], Twitter [40], etc.) and the COVID-19 trends has been investigated and analyzed in many different countries and regions, including specific studies in China [39], Europe [41], Canada [42], Italy [43], and the U.S. [44]. However, these articles focused on analyzing correlations and conducting longitudinal analysis of online search behaviors in various regions in the world, by considering generic COVID-19-related and symptom-related queries. None of them examined in detail the importance of the search queries with a carefully designed large-scale data-driven query identification process or attempted to forecast COVID-19 incidences and future outbreaks by incorporating the aforementioned prediction techniques. While the majority of the existing studies proclaim the potential predictive power of internet search data in COVID-19 forecasting through the corresponding correlation analyses, only a few have built prediction models [45–53] to fully utilize and demonstrate the predictive power of internet search data. This review study examines each of those prediction models, by elaborating the relevant internet search data as well as other alternative data, and analyzing the prediction models.

### 2.2. Search Strategy and Selection Criteria

In this section, the process to identify the articles for this review study is presented. Since this review study focuses on COVID-19 forecasting/prediction literature that utilizes internet search information, two constraints were imposed on the scope of this systematic and quantitative review study:

- Forecasting studies of COVID-19: papers that provided future predictions/forecasts for a specific region in the world and for a specific future horizon. The search terms used were as follows: COVID-19, coronavirus, SARS-CoV-2, prediction models, forecasting models, predictive analysis.
- Data-driven including internet search: broadly defined as papers that incorporated COVID-19-related data, internet search data, and other exogenous information into the setup or fitting of the model. Here, the internet search information was broadly defined as datasets that reflected the online search behaviors of a population of interest. The search terms used were as follows: internet search data, internet search information, Google Trends, online search behavior, COVID-19 time series information, mobility data.

Many different search term combinations, selected from the above imposed constraints, were used to retrieve desired research studies in online databases, including Google Scholar [54], Scopus [55], and PubMed [56]. Some of the search strings used were as follows: "COVID-19 prediction models" AND "internet search data"; "COVID-19 forecasting models" AND "Google Trends"; "COVID-19 prediction" AND "internet search information" AND "mobility data"; "COVID-19 forecast" AND "online search behavior" AND "time series information". After the initial search, 217 documents were retrieved, of which 28 were duplicates. Then, by further filtering with both constraints above, 9 studies were finally selected for this review.

Table 1 lists all the research studies considered in this paper, and their high-level overview of objectives, model types, data source, and quality assessments.

**Table 1.** Studies reviewed in detail in this article. List of abbreviations: COVID-19, coronavirus disease 2019; LASSO, least absolute shrinkage and selection operator; RMSE, root mean square error; MAE, mean absolute error; MAPE, mean absolute percentage error; correlation, Pearson correlation; persistence (model), rule-based baseline model that uses the count "today" as an estimate of all future predictions; AR, autoregressive model; LSTM, long-short term memory; ARIMA, autoregressive integrated moving average; PCA, principle component analysis; hospitalization, hospital admission; CI, confidence interval; PI, prediction interval; CDC, Center for Disease Control and Prevention; %ILI, percentage of influenza-like illnesses; Media Cloud, an open-source platform for media analysis, which shows the number of digital news articles covering the topic of interest in time series; health bot, data generated by virtual AI-based triage systems from Azure Microsoft; the output is a time series of number of people "flagged" with COVID-19.

| Study | Objective | Type of Model | Online Search Data Source | Other Data Inputs | Quality Assessment |
|---|---|---|---|---|---|
| Liu et al. (2020) [45] | Forecast 2-day-ahead COVID-19 cases in all 32 China provinces. | LASSO | Baidu | China CDC, Mobility, Media Cloud, COVID-19 cases | RMSE, Correlation: outperform persistence and AR baseline models in both metrics |
| Ayyoubzadeh et al. (2020) [46] | Forecast 1-day-ahead COVID-19 confirmed cases in Iran | Linear regression, LSTM | Google Trends | COVID-19 cases | RMSE: linear regression with Google search data performs better than LSTM with Google search data |
| Prasanth et al. (2021) [47] | Forecast 1-week-ahead COVID-19 cases and deaths in U.S., U.K., and India | LSTM | Google Trends | COVID-19 cases and deaths | RMSE, MAPE: LSTM has significant reduction from ARIMA baseline model |
| Rabiolo et al. (2021) [48] | Investigate the relationship between Google Trends symptom searches and COVID-19 cases and deaths; use ARIMA to predict COVID-19 cases and deaths in Australia, Brazil, France, Iran, India, Italy, South Africa, U.K., and U.S. up to 14 days ahead | PCA and ARIMA | Google Trends | COVID-19 cases and deaths | RMSE: models with search terms and COVID-19 time series information outperform those without |
| Lampos (2021) [49] | Forecast 1- and 2-week-ahead COVID-19 deaths in U.S., U.K., Australia, Canada, France, Greece, and South Africa. Produce point estimates and CI | Gaussian process (GP) | Google Trends | Media Cloud, COVID-19 deaths | MAE: the inclusion of search queries in GP autoregressive model significantly improves its performance |
| Turk et al. (2021) [50] | 14-day-ahead COVID-19 hospitalizations forecast in Greater Charlotte market area in U.S. | Vector autoregression (variant) | Google Trends | Mobility, health bot, COVID-19 hospitalizations | MAPE: outperform ARIMA (time series benchmark) model |
| Ma and Yang (2022) [51] | Forecast 1–4-week-ahead COVID-19 deaths in U.S. national and states level. Produce point estimates and CI | LASSO, spatialtemporal statistical approach | Google Trends | COVID-19 cases and deaths | RMSE, MAE, Correlation: outperform persistence and time series benchmarks, and perform reasonably against other CDC Forecast Hub methods |
| Wang et al. (2022) [52] | Forecast 1–2-week-ahead COVID-19 hospital admissions in U.S. national and states level. Produce point estimates | LASSO, spatialtemporal statistical approach | Google Trends | COVID-19 cases, vaccination rate | RMSE, MAE, Correlation: outperform persistence and time series benchmarks, and perform reasonably against other CDC Forecast Hub methods |

**Table 1.** *Cont.*

| Study | Objective | Type of Model | Online Search Data Source | Other Data Inputs | Quality Assessment |
|-------|-----------|---------------|---------------------------|-------------------|--------------------|
| Ma et al. (2022) [53] | Forecast 1–4-week-ahead COVID-19 cases, deaths, and 1-week ahead influenza. Produce point estimates and PI | LASSO, spatialtemporal statistical approach | Google Trends | COVID-19 cases, deaths, %ILI | RMSE, MAE, Correlation: outperform persistence and time series benchmarks, and perform reasonably against other CDC Forecast Hub methods |

## 3. Data Acquisition and Preprocessing

The literature considered in this paper (Table 1) focuses on various regions in the world. This section lists all the COVID-19-related data (forecasting targets), and all data inputs considered in the above studies, with the focus on internet search data. Details of data usage and availability of the selected research studies (Table 1) are further provided in Table S4 in Supplementary Materials.

### 3.1. COVID-19-Related Data (Forecasting Target)

Since the COVID-19 initial outbreak and the continuous spread, many different organizations around the globe, including the Center for System Science and Engineering (CSSE) at John Hopkins University (JHU) [57], China CDC [58], and European Center for Disease Prevention and Control (ECDC) [59], have maintained COVID-19-related data such as the number of confirmed cases and deaths, to keep track of the spread of the epidemic. These data are available in various geographical resolutions (country-wise or region-wise). The research studies (Table 1) consider some or all the COVID-19-related data listed below in various geographical resolutions as forecasting targets, as well as input features used in the models to assist the forecasts. Tables S1 and S2, in Supplementary Materials, also showcase the sample dataset of COVID-19 cases, deaths, and hospitalizations.

- Confirmed cases: daily/weekly COVID-19 new confirmed cases (infections) time series in different geographical resolutions. For example, the many U.S. studies used the JHU CSSE COVID-19 dataset [57] as the official ground truth. Confirmed case counts in China were obtained from China CDC [58]. The confirmed case counts in other regions were obtained from ECDC [59].
- Reported deaths: daily/weekly COVID-19 newly reported deaths time series in different geographical resolution. The data sources were similar to confirmed cases time series above.
- Hospitalizations: COVID-19 hospitalizations generally refer to the number of daily/weekly newly admitted patients to the hospitals in various geographical resolutions that tested positive for COVID-19. Hospitalizations reflect the number of severe cases, and therefore keeping track of hospitalizations is strategic for policymakers as it allows predicting the potential saturation of the hospital systems, and helping local public health officials make timely decisions in allocation of healthcare resources, such as ventilators, ICU beds, personal protective equipment, personnel, etc. Hospitalization data is maintained by different organizations across different regions in the world. For example, the U.S. Department of Health and Human Services (HHS) [60] releases the ground truth information of new hospital admissions in the U.S.
- Vaccination rates: percentage of fully vaccinated population in daily/weekly frequency, reported by different health organizations in each region. For example, U.S. vaccination rates are reported by the CDC [61] with daily frequency.

### 3.2. Internet Search Data

At the inception of a disease outbreak, official data may be unavailable or crude due to lack of a mature surveillance system. Internet search data, in the meantime, serves as a

great auxiliary tool for monitoring the spread of disease, which reflects the development of the disease almost instantaneously. Internet search data generally refers to how frequently one query term is searched or mentioned on the internet. There are many sources of search frequencies available to the public, such as Google Trends [62], Baidu Index [63], etc. Google Trends, for example, provides estimated Google search frequency time series for a desired query term in the specified geographical resolution and time frame [62]. In particular, Google Trends employs a sampling technique that collects all raw Google search frequencies of the query of interest and calculates the proportion that contains this query. The Baidu Index exhibits a similar sampling mechanism, which returns the search frequencies of a query (in Chinese) of interest. Due to its wide geographical coverage and popularity, Google Trends is used by the majority of the forecasting studies considering internet search data. Alternative internet search data sources are considered in regions where the Google search engine data are not considered as a representative sample, such as Baidu [63]. Table S3 in Supplementary Materials also showcases the sample dataset obtained from Google Trends.

Despite the advantages of internet search, the queries obtained from the internet are abundant and most of them are unrelated to the relevant disease. Furthermore, the raw search frequency time series might be noisy. Therefore, mechanisms for selecting important search queries and de-noising the search frequencies are necessary. The research studies (Table 1) incorporated different data-driven approach for selection and pre-processing, which will be discussed in detail below.

### 3.2.1. Query Selection

As correlation between COVID-19-related internet search queries and COVID-19 trends are well-studied in the literature [38,44], the majority of the research studies (Table 1) selected the relevant queries based on correlation. By starting with a large pool of COVID-19-related search terms in Mandarin, Liu et al. [45] first conducted a correlation study between the search terms and COVID-19 case counts, and eventually collected the top three search terms with highest correlations in the daily search fraction from Baidu [63] ("COVID-19 symptoms", "how many degree is fever", and "symptoms of fever"). Similarly, Prasanth et al. [47], Rabiolo et al. [48], and Turk et al. [50] selected the list of terms based on medical expertise [64] and prior correlation studies [44], where the selected terms were a combination of general COVID-19-related terms such as "COVID-19", "coronavirus", and specific intervention and symptom-related terms such as "hand sanitizer", "mask", "cough", "fever", "shortness of breath", etc. Furthermore, as people tend to search for COVID-19-related information online before they arrive at a clinic or are tested positive [65], COVID-19-related search volumes tend to peak prior to reported cases or deaths. Several studies exploited this by computing the optimal delay between search frequencies and COVID-19 trends, to further select the important queries based on the association between optimally lagged search queries and COVID-19 trends. Lampos et al. [49] determined the list of search terms by COVID-19-related symptoms and keywords, and computed the time lags between the search frequencies and COVID-19 confirmed cases and deaths, as inputs to their forecasting model. Ma and Yang [51], Wang et al. [52], and Ma et al. [53] developed an end-to-end data-driven selection mechanism that selected 23 important search queries from the 256 top searched COVID-19-related Google search terms, by ranking the Pearson correlation coefficient between optimal lagged search term and COVID-19 trends (forecasting target), with a cutoff threshold of 0.5, using summer 2020 as the training period. The selected 23 important search terms contained more specific COVID-19-related symptoms, such as "loss of taste" and "loss of smell", compared to other studies. Figure 1 shows the delay in peaks between the Google search query, "loss of taste", and COVID-19 cases (a) and deaths (b) trend. It illustrates the association between COVID-19 trends and important search queries, as well as the optimal time lag in between, which could be helpful for early outbreak detections.

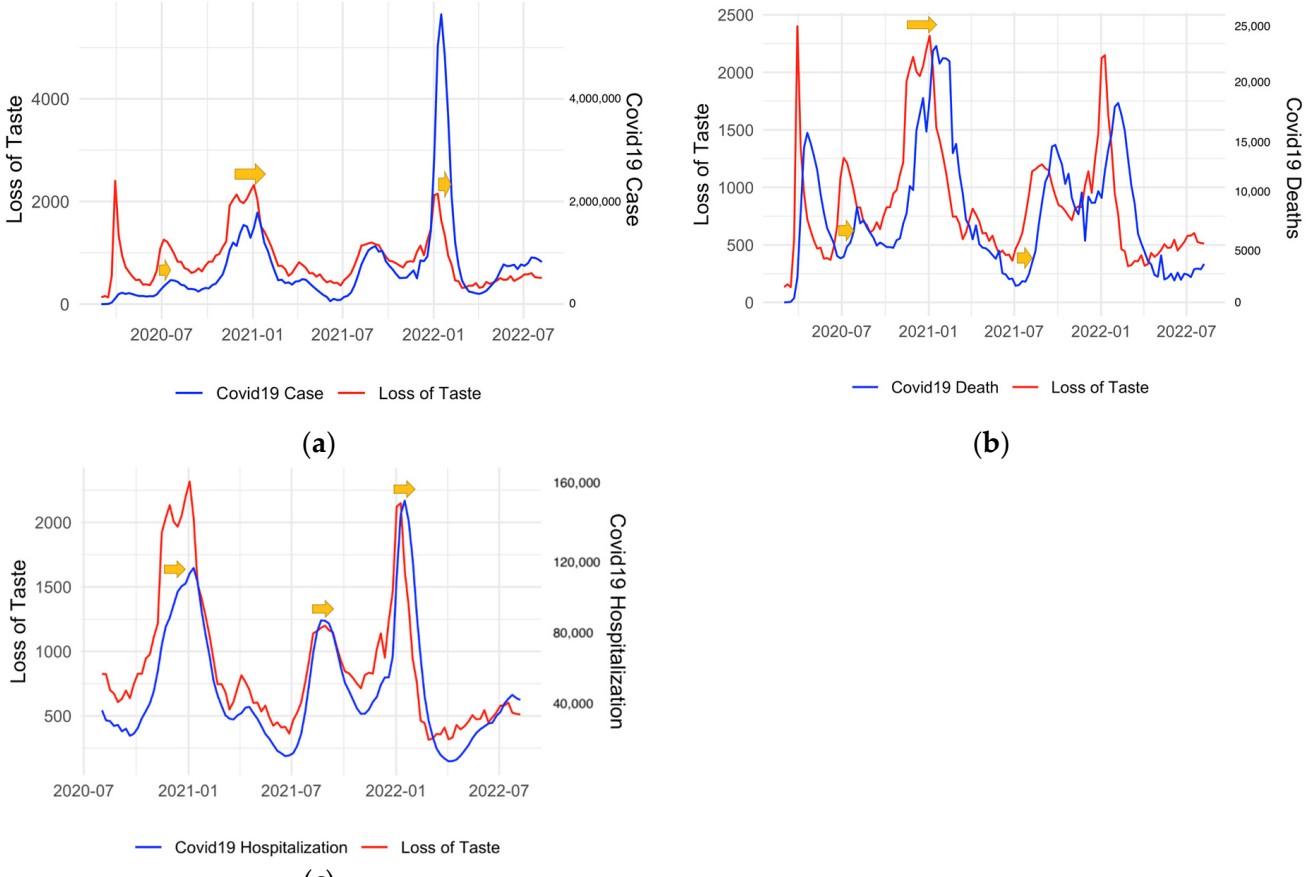

**Figure 1.** Google search query "loss of taste" [62] and United States national-level COVID-19 weekly cases [57] (**a**), deaths [57] (**b**), and hospitalizations [60] (**c**), from 1 March 2020 to 7 August 2022. The Google search query "loss of taste" search frequency [62] is in red. COVID-19 U.S. national level weekly cases [57], deaths [57], and hospitalizations [60] trends are in blue. Y-axes are adjusted accordingly. This figure illustrates the delay in peak between the search frequency of "loss of taste" and COVID-19 cases, deaths, and hospitalizations. The delay in peak is illustrated by the yellow arrow in the figure.

3.2.2. Search Volume Data Preprocessing

Being user-dependent by design, estimators based on internet search data embody many sources of uncertainty and instability. The frequencies of the obtained internet search queries from Google Trends are sparse and they have sudden spikes/drops due to natural noises in sampling approaches. In particular, Google Trends truncates data to 0 if the search volume for the query is too low. Thus, the zeros in a particular query's search frequencies obtained from Google Trends indicate missing data due to low search volume, which is very common in practice, especially for low-density population areas and low-internet-connected areas. Moreover, search queries obtained from Google Trends could have sudden spikes/drops due to natural noises in its sampling approaches. Researchers have developed different data-driven techniques to filter the selected search queries' frequencies, before employing them in forecasting models. Liu et al. [45] employed a 2-day moving average in all the selected queries to enhance signal and reduce noise, while Rabiolo et al. [48] conducted principle component analysis (PCA) and used two principle components from all the selected queries for subsequent forecasts. On the other hand, Ma and Yang [51], Wang et al. [52], and Ma et al. [53] first addressed the high sparsity in the state-level data through a weighted average of state-level and regional or national-level search queries' frequencies, and incorporated an interquantile range (IQR)-based data filtering mechanism to further smooth out extreme outliers due to sudden spikes/drops. Wang et al. [52] further

applied a 7-day moving average to remove the weekly seasonality in the selected Google search queries, before employing them in the hospitalizations forecasts. Lampos et al. [49] took a step further by not only filtering out the noise in the search queries' frequencies but also the news media effects led by public fear. They proposed an autoregression (AR) model that minimizes the news media effects on the search queries, by splitting the search population into infected and concerned. By estimating the infected population within the search queries' frequencies, they demonstrated the robustness of search queries serving as early warning signals for future waves, and significantly improved the forecasting performances in all regions of interest.

*3.3. Other Auxiliary Data Sources*

Besides COVID-19-related data serving as time series information and internet search data serving as exogenous variables, the research studies (Table 1) also incorporated other auxiliary data in their forecasting frameworks.

News media reports, referring to time series of numbers of COVID-19-related news media articles, also contain strong associations with COVID-19 trends as exogenous variables [45]. Media Cloud is an online open-source platform that contains news media reports. By specifying the keywords, Media Cloud allows the tracking of media on a particular topic or multiple topics of interest related to the keywords, and is used in many of the forecasting studies above. Media Cloud will return a time series of the number of related news articles available over time from a collection of media websites using the specified keywords, which represents the region-specific news/media activity trends. Liu et al. [45] treated this as exogenous time series information as an input of their forecasting model. On the other hand, Lampos et al. [49] used media coverage for preprocessing of the selected search queries, in order to minimize the news media effect on search queries' frequencies.

Mobility data have also been shown to assist COVID-19 forecasts, serving as an exogenous variable [66,67]. Google mobility [68], Apple mobility [69], and Facebook movement range maps [70] are the most popular mobility data sources used in the literature studies. They all share a similar structure, where the reports record a community's daily movement in different areas such as retail and recreation, groceries and pharmacies, parks, transit stations, workplaces, and residential addresses. A baseline mobility index is established for 7 days in a week, which represents the usual community mobility value. The mobility data reports the movement pattern of the community, with the percentage of changes from the baseline day's value. Turk et al. [50] incorporated both Apple mobility reports and Facebook movement range maps in their forecasting model, in addition to Google search data and COVID-19 time series information.

As the COVID-19 pandemic continues, severe seasonal influenza may break out alongside COVID-19, causing additional burdens on healthcare resources and public safety. Influenza outbreaks happening alongside the COVID-19 wave could also potentially serve as useful information for COVID-19 forecasts, and vise-versa. For decades, the U.S. CDC has monitored flu activities through the Influenza-like Illness Surveillance Network (ILINet), which collects the weekly reported number of outpatients with influenza-like illness (ILI) from thousands of healthcare providers and publishes the weekly ILI percentages. Ma et al. [53] proposed an accurate COVID-19 and influenza joint forecasting framework, ARGOX-Joint-Ensemble, which efficiently incorporates previously proposed ILI and COVID-19 forecasting models into a new ensemble framework that pools the information between influenza and COVID-19. In particular, it incorporates real-time influenza information for COVID-19 cases, deaths, and hospitalizations forecasts, and vise-versa for ILI forecasts.

## 4. Methods

The selected papers (Table 1) aim to forecast the spread of COVID-19 with different models, harnessing the predictive power of internet search data and other auxiliary data source introduced above. In general, the prediction models used are data-driven statistical

and machine learning techniques. This section briefly introduces the various prediction models used by the 9 identified studies (Table 1), as well as the subsequent evaluation process for the forecasting results. Details of model implementations and code availabilities of the selected studies are further provided in Table S4 in Supplementary Materials.

*4.1. Prediction Models*

4.1.1. Statistical Models

Incorporating information up to the current time point to generate future predictions with statistical and probabilistic properties, statistical models are widely used in disease forecasting [12–15], and can be generally categorized into linear and nonlinear-based models.

Linear-based models are mostly connected with regression models but can be also referred to models assuming linear connection between a response variable (target) and explanatory variables (exogenous information). In the disease forecasting task, one of the most used linear-based models is the autoregressive moving average model (ARMA), and its variants, such as AR, MA, ARIMA, etc. ARMA models predict future data in a series using past data, by representing the relationship between past, current, and future values of the same time series as a linear function [71]. Their simplicity of structure provides flexibility of incorporating additional exogenous variables (ARMA-X and other variates), and strong interpretability of autocorrelation and the relationship with all exogenous variables incorporated. Rabiolo et al. [48] fitted an ARIMA model to forecast COVID-19 cases and deaths in multiple countries by treating Google search data (PCA dimensionality reduced) as exogenous variables, and demonstrate that a simple ARIMA model can perform reasonably in forecasting the COVID-19 pandemic. Meanwhile, Turk et al. [50] used a vector-autoregression (VAR)-based model, combining Google search data, healthcare chatbot scores, and mobility information, to produce COVID-19 hospitalizations predictions in the Greater Charlotte market area. In addition, one can incorporate $L_1$-norm penalty into an ARMA structure to achieve model selection while enjoying the interpretability and simplicity from the linear structure. Ma and Yang [51], Wang et al. [52], and Ma et al. [53] proposed an autoregression-like model that combines COVID-19 time series information and Google search data with $L_1$-norm penalty for U.S. national COVID-19 forecasts. They demonstrated that past COVID-19 time series and Google search information efficiently complement each other in a rolling-window forecasting manner. More detailed descriptions are presented in Section 2.2 in Supplementary Materials.

On the other hand, the targeted time series might exhibit a more complicated structure and relationship with itself, and the exogenous variables considered, that cannot be easily captured by linear-based models. A nonlinear model generally refers to empirical or semi-empirical modeling that takes at least some nonlinearities into account. An example of nonlinearity in COVID-19 forecasts is that the number of cases leads to a disproportionate number of deaths during the peak periods of a COVID-19 wave [2]. More technical details on nonlinearities in time series are discussed in [72]. The Gaussian process (GP), defined as a stochastic process such that every finite collection of those random variables has a multivariate normal distribution [73], is one of the nonlinear-based models that is commonly used in time series forecasting, and justified by prior works on modeling infectious diseases [74,75]. Lampos et al. [49] develop a GP-based forecasting model to provide D-day-ahead COVID-19 death predictions in multiple countries, and showcased how Google search data can enhance early warning signals compared to baseline models. Other nonlinear modeling attempts are also considered among the research studies. For instance, Ma and Yang [51] incorporated an ensemble framework that combined their U.S. state-level COVID-19 death predictions, generated from three different submodels, and selected the best predictor for each week for 1–4-week-ahead COVID-19 death forecasts.

### 4.1.2. Deep Learning Models

Deep learning methods, trained using supervised learning, are deep neural networks that learn directly from input data to predict some targets. Such models are more flexible than both mechanistic models and statistical models, due to their representation capability and less sophisticated handcrafting preprocessing of the input data. In time series forecasting problems, many deep learning-based models take inspiration from natural language processing applications, including long short-term memory (LSTM) [76], gated recurrent unit (GRU) [77], etc. All the above models can capture intrinsic information from sequential data for accurate prediction. LSTM models are a type of recurrent neural network (RNN) [78], and are able to overcome the drawbacks of the vanilla RNN, the exploding and diminishing gradients problem, by additionally "remembering" portions of the past, so-called memory [76]. Therefore, LSTMs are capable of learning and capturing sophisticated relationships between the target and input data with long time lags in between, and produce robust and accurate forecasts. Further details of the vanilla LSTM structure are demonstrated in Supplementary Materials Section 2.1 and Figure S1. Prasanth et al. [47] used the search data from selected queries on a baseline LSTM structure for U.S, U.K, and India 1-week-ahead COVID-19 cases and deaths predictions. Ayyoubzadeh et al. [46], on the other hand, fitted a linear regression model and a vanilla LSTM model, both incorporating Google search data, to provide short-term a COVID-19 cases forecast in Iran, and compare the performances with each other.

### 4.2. Evaluation Process and Metrics

The selected papers evaluate the forecasts, produced from the developed prediction models, through comparison against alternative benchmarks via different error metrics. To evaluate the accuracy of a point estimate of COVID-19 target (cases, deaths, and hospitalizations) against the actual ground truth, the selected studies used several different evaluation metrics, including root mean squared error (RMSE), mean absolute error (MAE), mean absolute percentage error (MAPE), and Pearson correlation (correlation). RMSE between an estimate $\hat{y}_t$ and the true value $y_t$ over period $t = 1, \ldots, T$ is $\sqrt{\frac{1}{T} \sum_{t=1}^{T} (\hat{y}_t - y_t)^2}$. MAE between an estimate $\hat{y}_t$ and the true value $y_t$ over period $t = 1, \ldots, T$ is $\frac{1}{T} \sum_{t=1}^{T} |\hat{y}_t - y_t|$. MAPE between an estimate $\hat{y}_t$ and the true value $y_t$ over period $t = 1, \ldots, T$ is $\frac{1}{T} \sum_{t=1}^{T} \left| \frac{\hat{y}_t - y_t}{y_t} \right|$. Correlation is the Pearson correlation coefficient between $\hat{y} = (\hat{y}_1, \ldots, \hat{y}_T)$ and $y = (y_1, \ldots, y_T)$. To evaluate the accuracy of a probabilistic prediction and confidence or prediction interval of COVID-19 target (cases, deaths, and hospitalizations) against the actual ground truth, the selected studies used two different evaluation metrics: the weighted interval score (WIS) and the empirical coverage. The weighted interval score (WIS) [79] is a proper scoring rule (smaller is better), that takes an entire predictive distribution into account and penalizes over- and under-confidence. Following CDC Forecast Hub's submission guideline [5] in this study, the WIS between the true value and a predictive distribution at time was evaluated across 11 prediction intervals with nominal coverages of 98%, 95%, 90%, ..., 10%. The WIS between the true value $y = (y_1, \ldots, y_T)$ and predictive distributions over period $t = 1, \ldots, T$ was computed by averaging the WIS across the period. The empirical coverage between the true value $y = (y_1, \ldots, y_T)$ and the prediction intervals over period $t = 1, \ldots, T$ is the proportion of true values falling inside a given central prediction interval with 95% nominal coverage.

The evaluation process is generally performed as follows: (1) the point estimates (or probabilistic predictions) of a particular COVID-19 target in a specific region and time interval of interest are produced from the developed model; (2) a particular evaluation error metric is chosen and the corresponding error is computed; (3) alternative benchmark point estimates (or probabilistic predictions) of the same COVID-19 target in the same region and time interval are collected; (4) benchmark error is computed with the same evaluation error metric; (5) performance analysis is conducted by comparing the error reduction between the developed model and benchmark.

By choosing the appropriate benchmark, one can directly evaluate the prediction performances of COVID-19 waves in different regions and different periods, focusing on point estimates, while further interpreting the uncertainties in different rapidly changing dynamics, focusing on probabilistic predictions. The comparisons against different benchmarks could also provide additional insights on the importance of a particular exogenous data and model structures of the developed forecasting framework. More guidelines on COVID-19 forecast reporting and evaluation metrics can be found in [80,81].

**5. Results**

With the foundations of prior correlation analysis between internet search data and COVID-19 trends, the selected research studies (Table 1) took a step further by systematically uncovering potentially useful online search queries, filter/de-noising, and utilizing them to forecast COVID-19 trends. They illustrate the predictive power of public search information in their proposed forecasting models, by producing COVID-19 predictions in the geographical regions and the time frames of interest. Finally, they evaluated the forecasts through different evaluation metrics, shown in Section 4.2. Generally, the selected studies analyzed their models' predictive power, by comparing the model forecasts with different benchmark forecasts, briefly summarized below.

- Persistence (naïve) rule: a rule-based model that uses the COVID-19 target (cases, death, or hospitalization) count at date T as an estimate of the prediction for $T + \delta_t$.
- Time series baseline: generally refers to linear-based models such as the autoregressive moving average model (ARMA) [71], and its variants (AR, MA, ARIMA, etc.), that utilize COVID-19-related time series information only (in Section 3.1).
- Simpler version of proposed model: generally refers to a simpler version of the proposed forecasting model after removing one or multiple components in the model structure (data component, architecture component, etc.).
- Other publicly available benchmark: generally refers to established and publicly available benchmark predictions, such as those of the COVID-19 forecast hub [5].

*5.1. Importance of Internet Search Component*

Although the forecasting targets, regions, time frames, and benchmark comparisons may differ from among all considered studies, they all demonstrate the importance of the internet search data component in their forecasting models through detailed sensitivity and evaluation analysis. In general, the sensitivity analysis was conducted by comparing the proposed model forecasts against other alternative benchmark forecasts, which had a similar model structure but did not take internet search data into account. A few of the selected studies took a step further to compare data against benchmarks with an alternative model structure, to further illustrate the robustness and accuracy achieved with the internet search data.

Focusing on 51 U.S. states/districts (including Washington D.C.) and U.S. national level, Ma and Yang [51] produced 1–4-week-ahead COVID-19 death predictions from 4 July 2020 to 5 March 2022, using the developed data-driven spatiotemporal statistical framework with $L_1$-norm penalty and incorporating Google search queries obtained in different geographical resolutions. They compared their proposed method with time series baseline models (without Google search information) and found that the Google search information contributed to around 22% RMSE and 27% MAE error reductions at the national level, demonstrating the additional predictive power of internet search data. Meanwhile, on the state level, they compared their proposed state-level model with simpler versions of the proposed models by removing the spatiotemporal and ensemble model structure, and achieved around 18% RMSE and 20% MAE error reductions. They further showcased the coverage of the state-level forecast confidence intervals, where they reasonably measured the accuracy of their weekly estimates, albeit with overconfidence. These indicate that additional ensemble (nonlinear) and spatiotemporal model structures can further enhance the predictive power of Google search data. The detailed sensitiv-

ity analysis on the state level demonstrates that the internet search data's noisiness and sparsity can be minimized, and its predictive power can be maximized with principled and data-driven model structures. They also show competitive performance with other publicly available COVID-19 forecasts from other research teams making predictions of COVID-19 deaths [5]. Wang et al. [52] adapted a similar framework and extended internet search data's predictive power to 2-week-ahead COVID-19 hospitalization predictions at both U.S. national and state levels. By comparison against an autoregressive model, their proposed model achieved 18% and 25% reductions in RMSE and MAE at the national level, respectively, while achieving roughly 8% and 12% reductions in RMSE and MAE on average at the state level, respectively. The significant improvements from time series benchmark predictions further emphasizes the importance of internet search data alongside COVID-19 time series information. Furthermore, their proposed ARGO-inspired method yielded roughly 35% and 12% RMSE error reductions for 1- and 2-week-ahead state-level forecasts, compared with two benchmark models published by the COVID-19 forecast hub [5], emphasizing the efficient combination of Google search information and COVID-19 time series information in different rapidly changing dynamics. Ma et al. [53] took a step further by combining prior (COVID-19 and influenza) single disease forecasting frameworks into joint ensemble forecasts for 1–4-week-ahead COVID-19 cases and deaths, and 1-week-ahead influenza-like illnesses. The proposed framework improved upon previous single disease forecasts, and significantly outperformed alternative time series benchmarks and persistence models, in both point estimates and probabilistic predictions (prediction intervals) in all the error metrics in Section 4.2, further illustrating Google search data's additive predictive power for both infectious diseases. The proposed bi-disease prediction model also remains competitive against other publicly available forecasts. Meanwhile, Turk et al. [50] used their proposed vector-autoregression (VAR)-based model to produce 14-day-ahead out-of-sample COVID-19 hospitalization predictions in the U.S. Greater Charlotte market area from 2 August 2020 to 15 August 2020, incorporating both Google search data and mobility information. Their proposed VAR-based model achieved around 22% MAPE error reduction, compared to the baseline ARIMA, showing that efficiently combining internet search data with time series information can overcome lagging behaviors in forecasts, while avoiding overshooting and underestimations. Lastly, Prasanth et al. [47], Rabiolo et al. [48], and Lampos et al. [49] also produced U.S. national level COVID-19 forecasts, illustrating the strength of internet search data serving as early warning signals from different angles.

Some of the selected research studies (Table 1) also focused on other regions and countries in the world. Liu et al. [45] produced 2-day-ahead and real-time COVID-19 cases forecasts for 32 Chinese provinces for the time period spanning from 3 February 2020 to 21 February 2020. Their proposed ARGONet + GLEAM model outperformed both persistence and AR models in 27 out of 32 Chinese provinces, and produced reasonable disease estimates in the rest of the provinces, showcasing public search behavior as a valuable alternative data source assisting short-term forecasts when the amounts of data during the early stage of an emerging outbreak are limited. Ayyoubzadeh et al. [46] predicted 1-day-ahead COVID-19 cases in Iran from 15 February 2020 to 18 March 2020, using basic LSTM and linear regression models. They showed that a linear regression model with Google search information can achieve better performance when data are limited in the early stage of pandemic, compared to an LSTM model with Google search information. By looking at the feature/coefficient weights in both models, they found that most effective search queries were handwashing, hand sanitizer, and antiseptic topics, besides historical COVID-19 case counts. Similarly, Prasanth et al. [47] forecasted 1-week-ahead COVID-19 cases and deaths in India, the U.S. and U.K. during the period from 14 May 2020 to 20 May 2020, by training a hybrid grey wolf optimizer (GWO)-LSTM model from 24 February 2020 to 13 May 2020. By conducting sensitivity analysis after removing some of the features and simplifying model structures, their full model (with internet search data, time series information, and GWO) achieved roughly 55%, 4%, and 52% MAPE

error reductions compared to the best-performing alternative models in India, the U.S., and the U.K., respectively, for COVID-19 case predictions (similarly for death predictions). Rabiolo et al. [48] first conducted studies of correlation between selected internet search queries and COVID-19 growth trends in nine countries (Australia, Brazil, France, Iran, India, Italy, South Africa, the U.K., the U.S.), and predicted 14-day-ahead COVID-19 cases and deaths using ARIMA during the period from 22 January 2020 to 20 December 2020. Overall, predictions based on both search terms and COVID-19 conventional metrics performed better than those not including Google searches (13% RMSE error reduction on average), illustrating early warning signals provided by the selected queries during the first two COVID-19 waves in different countries. Lampos et al. [49] produced 1- and 2-week-ahead COVID-19 death point estimates and probabilistic predictions (confidence intervals) in the U.S., the U.K., Australia, Canada, France, Greece, and South Africa from 17 February 2020 to 24 May 2020, indicating that the inclusion of important search queries (with news media effect minimized) generally improved model accuracy compared with basic persistence models.

*5.2. Internet Search Information Serving as Early-Warning Signals*

Forecasting COVID-19 trends during the initial outbreak and subsequent waves has been extremely challenging, due to noisy and unreliable ground truth data, limited historical data, and most notably the evolving nature of COVID-19 outbreaks. Meanwhile, internet search data could serve as early-warning detection signals for future outbreaks, and "foresee" surges of upcoming COVID-19 waves (Figure 1). Liu et al. [45] showed that historical COVID-19 confirmed cases and internet-based search terms from Baidu were consistently relevant sources of information over most of the study period. With evaluations of the parameter weights, the importance of media article counts decreased over time, whereas internet-based search terms retained their importance. Furthermore, by incorporating the internet search terms, the proposed model overcame lagging behaviors in its forecasts in the majority of the provinces compared to the time series benchmarks. Although online searches can be driven by concern rather than infection, especially in the early stage of the COVID-19 pandemic, Lampos et al. [49] showed that after incorporating news media coverage to minimize this effect, the output from the proposed model provided useful insights, including early warnings for potential disease spread, and showcased the effect of physical distancing measures. Through thorough correlation studies, Rabiolo et al. [48] observed that the Google searches of COVID-19 symptoms exhibited high similarities with COVID-19 trends with certain time lags, similar to Figure 1. This behavior can significantly contribute to the early warning of new waves and surges. Ayyoubzadeh et al. [46], Prasanth et al. [47], and Turk et al. [50] also demonstrated such behaviors during the early stage of the pandemic as well. Meanwhile, by examining the proposed model during three periods with different rapidly changing dynamics in the U.S. (COVID-19 second wave, COVID-19 Delta variant, and COVID-19 Omicron variant), Ma et al. [53] demonstrated the predictive power of Google search data serving as early-warning signals, as the proposed model produced robust early-warning estimates before the increasing and peaking periods and was less prone to overestimation in all three periods considered.

## 6. Discussion

Based on the intuition that COVID-19-related keyword search frequencies reflect, to an extent, the number of people presenting symptoms related to COVID-19 before their arrival at a clinic, these studies conducted end-to-end COVID-19 forecasting tasks during different COVID-19 waves. By recognizing that COVID-19-related search queries follow a similar trend to that of the COVID-19 epidemic and precede traditional COVID-19 metrics (Figure 1), the research studies incorporated the internet search data efficiently in their proposed models to allow early recognition of new waves and epidemic peaks, potentially assisting governments and healthcare organizations to prepare for the newly infected cases, and allocate hospital resource. The robustness and accuracy of COVID-19 forecasts, shown

in all the selected research studies (Table 1), demonstrate that internet search data have great potential in assisting short-term infectious disease forecasts.

However, big data-driven forecasting methods also have limitations. One of the limitations is that the internet search data are sensitive to media coverage and the inherit sampling noise, and such instability could propagate into the COVID-19 predictions. By conducting thorough correlation studies in various countries, Lampos et al. [49] discovered that public search behavior could signal the presence of actual infections. However, this could also be attributed or inflated by general concern, intensified by news media coverage, increased mortality across the world, and imposed physical distancing measures, especially before subsequent outbreaks led by different COVID-19 variants. Ma and Yang [51] and Wang et al. [52] realized that Google search data can be noisy due to the instability of Google Trends' sampling approach. Especially for state-level Google search data, the lack of search intensity can make the search data unrepresentative of the real interest of the people. To combat these challenges, Lampos et al. [49] proposed an AR-based model to minimize the news media effects in the search queries, by stripping out the concerned population from the "infected" population, and inputting the preprocessed queries into their nonlinear forecasting model for multiple country death forecasts. Meanwhile, Ma and Yang [51] and Wang et al. [52] took a different approach by proposing an IQR-based data-preprocessing framework, and selected the most important search queries (based on correlation) for subsequent forecasting models. Wang et al. [52] further applied moving average smoothing to Google search data and used national level search frequencies directly as input features for state-level predictions to further account for instability in the state-level Google search queries. Liu et al. [45] also applied 2-day moving average smoothing to case counts, search volumes, and media articles (Media Cloud) to enhance signal and reduce noise.

Additionally, information in internet search data deteriorates as forecast horizons expand, which could potentially impact long-term forecasting performances. Ma and Yang [51] and Wang et al. [52] realized the deterioration of accuracy in their 4-week-ahead COVID-19 death forecasts, and 3–4-week-ahead COVID-19 hospitalization forecasts. Liu et al. [45] showed that the limited amount of epidemiological and internet search information constrained their capacity to produce reliable long-term forecasts during the initial COVID-19 outbreaks. Similarly, Turk et al. [50] also stated that the limited hospitalization data during early stages of the pandemic restricted them in producing more geographically granular estimates. Nevertheless, by modifying their proposed model structure (incorporating $L_1$-norm penalty [51,52]), including additional features (health bot [50] and COVID-19 time series information [51,52]), and adjusting the forecasting horizon in different geographical resolutions [45,46,48], they were able to compensate this limitation and still do better than baseline benchmark models (in their results analysis). Models to further alleviate the bias in internet search data and capture long-term COVID-19 trends could be an interesting future direction.

Despite the limitations, big data is proven to be a valuable resource for infectious disease forecasts, which can significantly boost traditional statistical models and deep learning method forecasting performances. Across different forecasting targets, different regions, and different time spans, these studies provide comprehensive analysis on using online search data as an early indicator of COVID-19 under different COVID-19 variants, potentially assisting healthcare officials and promoting general public awareness. Furthermore, as internet search information has been shown to track successfully with various diseases such as influenza [12,16,18,22], dengue [17], and Zika [82,83], among others [84], the selected research studies in this paper provide additional applicability and predictive insights for internet search data, and take a step further in prediction of future infectious diseases.

Many future directions remain open for research on internet search data applicability and infectious disease modeling, beyond addressing the current limitations stated above. On the data modeling front, combining mechanistic models with data-driven statistical or deep learning models could be an interesting future direction. Examples include physics-

informed neural networks with ordinary differential equation constraints [85], among others. On the disease dynamics front, as the COVID-19 pandemic continues, different infectious diseases may break out alongside COVID-19 in different regions, including severe seasonal influenza [86] and monkeypox [87], causing additional burdens on healthcare resources and public safety. Accurate and real-time joint prediction of various infectious diseases in different geographical resolutions in a reliable and timely manner is an urgent and interesting future direction.

## 7. Conclusions

The researchers have studied different data-driven query selection/filtering mechanisms and predictive models to harness the association between internet search data and COVID-19 trends, published in [45–53]. They emphasized that the success of the forecasting models is because COVID-19-related public search behavior has great potential to uncover upcoming disease outbreaks, serving as a useful external signal to baseline time series forecasting models. The outputs from all the proposed models provide useful insights, including early warnings for potential disease spread [45–53], and showcase the effect of different healthcare interventions implemented by government officials, such as physical distancing measures [49] and vaccination [52]. By comparing the results with benchmark models, the research studies illustrate that signals from web search data could have served as preliminary early indicators for COVID-19 prevalence across various regions at different geographical resolutions. The immediate impacts of various government interventions can also be measured via these forecasting models, and can be illustrated in the internet search queries' frequencies. Lastly, the qualitative analysis and correlation studies show that COVID-19-specific symptoms or COVID-19-related generic queries correlate better with, and are more predictive of, clinically reported metrics. Overall, the research studies investigated in this review paper illustrate the predictive power of online search data in infectious disease forecasts across different targets, regions, and time ranges, demonstrating robustness and serving as strong external signals in traditional disease tracking frameworks.

**Supplementary Materials:** The following supporting information can be downloaded at: https://www.mdpi.com/article/10.3390/analytics1020014/s1, Table S1: Example COVID-19 new cases and deaths dataset; Table S2: Example COVID-19 hospitalization dataset; Table S3: Example Google Trends' dataset; Table S4: Data and Code availabilities of the selected research studies in this review paper; Figure S1: Illustration of the architecture of a baseline LSTM cell.

**Author Contributions:** Conceptualization, S.M., Y.S. and S.Y.; formal analysis, S.M.; investigation, S.M.; writing—original draft preparation, S.M. and S.Y.; writing—review and editing, S.M., Y.S. and S.Y.; supervision, S.Y.; project administration, S.Y.; funding acquisition, S.Y. All authors have read and agreed to the published version of the manuscript.

**Funding:** S.Y.'s research is partly supported by the National Center for Advancing Translational Sciences of the National Institutes of Health under Award number UL1TR002378. The content is solely the responsibility of the authors and does not necessarily represent the official views of the National Institutes of Health.

**Institutional Review Board Statement:** Not applicable.

**Informed Consent Statement:** Not applicable.

**Data Availability Statement:** Not applicable.

**Conflicts of Interest:** The authors declare no conflict of interest.

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
