# Peer review of "Using Internet Search Data to Forecast COVID-19 Trends: A Systematic Review"

_2813-2203, doi:10.3390/analytics1020014_

Round 1
Reviewer 1 Report
1 Showcasing the sample dataset is important.
2. Elaborate the process of evaluation in the methods section.
3. Elaborate on the deep learning models.
4. Results should be better explained to show to the research community about the the outcomes of the study.
5. Results should be more presentable for better understanding.
Author Response
Summary of Changes in this Revision:
- We have re-organized the Results section, according to the suggestion from reviewer #1.
- We have an additional section 2.2 to briefly describe the search strategy and selection criteria of the studies considered in this review paper, according to the suggestion from reviewer #3. All details in the main manuscript are updated accordingly.
- We have added more detailed descriptions for deep learning models and statistical models in section 4, and Supplementary Materials section 2, according to the suggestions from reviewer #1 and reviewer #1.
- We have moved the evaluation procedures of forecasting models to an independent section, section 4.2, and added more descriptions on each error metrics, according to the suggestion from reviewer #1 and #3. All details in the main manuscript are updated accordingly.
- We have revised the section 5 Results, and re-organized the section for clarity, according to the suggestions from reviewer #1. All details in the main manuscript are updated accordingly.
- We have added COVID-19 targets’ sample dataset in the Supplementary Materials section 1, according to the suggestion from reviewer #1 and #2. All details in the main manuscript are updated accordingly.
- We have added additional descriptions for contributions of this review study and its viability in reapplying the methods in the Discussion section and Supplementary Materials section 3, according to the suggestion from reviewer #4.
- We have added additional future directions discussion in the Discussion section, according to the suggestions from reviewer #2.
Replies to Comments Provided by Reviewer #1
Response: We thank the reviewer for all the suggestions. The detailed comments greatly improve the clarity of the manuscript. We have revised the paper accordingly, and all the changes are written in color blue.
Comment: Showcasing the sample dataset is important.
Response: We thank the reviewer for pointing out the lack of illustration for the COVID-19 data. We have now showcased sample datasets of COVID-19 cases, deaths and hospitalization in selected countries and dates for clearer illustration of the COVID-19 targets, in the Supplementary Materials. Table S1 showcases the daily COVID-19 new case and death counts in multiple countries (considered in the forecasting studies in Table 1), which are obtained from JHU CSSE COVID-19 dataset [56]. Table S2 showcases the daily COVID-19 hospitalizations in the United States, which are obtained from the U.S. HHS [59]. All other countries’ cases, deaths and hospitalizations follow the same structure as shown in Table S1 and S2. As an example, “standing on” 2022-08-07, to produce 1-4 weeks ahead forecasts (2022-08-08 to 2022-09-04), one can use COVID-19 cases and deaths information up to 2022-08-07. We have also edited section 3.1, referencing Table S1 and S2 as sample datasets, which is quoted as follow:
“Table S1 and S2, in Supplementary Materials, also showcases the sample dataset of COVID-19 cases, deaths and hospitalizations.”
We have also added table S3 in the Supplementary Materials to showcase the internet search dataset. Specifically, Table S3 displays the search frequencies of “Loss of Taste”, obtained in daily frequencies in two U.S. states. Other time frames and geographical resolutions can be specified when retrieving a query of interest. We have added the following sentence in section 3.2 to refer to Supplementary Materials:
“Table S3 in Supplementary Materials also showcases the sample dataset obtained from Google Trends.”
Comment: Elaborate the process of evaluation in the methods section.
Response: We thank the reviewer for the suggestion on elaborating the forecasts’ evaluation process. We have now moved the evaluation discussions to a separate section, section 4.2, and added probabilistic predictions’ error metrics, discussions on the evaluation process and references on COVID-19 forecasts’ evaluation guidelines. The revised section is also quoted below:
“To evaluate the accuracy of a probabilistic prediction and confidence or prediction interval of COVID-19 target (cases, deaths, and hospitalizations) against the actual groundtruth, the selected studies use two different evaluation metrics:the weighted interval score (WIS) and the empirical coverage. The weighted interval score (WIS) [79] is a proper scoring rule (smaller is better), that takes an entire predictive distribution into account and penalizes over- and under-confidence. Following CDC Forecast Hub's submission guideline [11] in this study, the WIS between the true value and a predictive distribution at time is evaluated across 11 prediction intervals with nominal coverages of 98%, 95%, 90%, …, 10%. The WIS between the true value and predictive distributions over the period t=1… T is computed by averaging the WIS across the period. The empirical coverage between the true value and the prediction intervals over period t=1…T is the proportion of true values falling inside a given central prediction interval with 95% nominal coverage.
The evaluation process is generally performed as follows: (1) the point estimates (or probabilistic predictions) of a particular COVID-19 target in a specific region and time interval of interest is produced from the developed model; (2) A particular evaluation error metric is chosen and the corresponding error is computed; (3) alternative benchmark point estimates (or probabilistic predictions) of the same COVID-19 target in the same region and time interval is collected; (4) benchmark error is computed with the same evaluation error metric; (5) performance analysis is conducted by comparing the error reduction between developed model and benchmark.
By choosing the appropriate benchmark, one can directly evaluate the prediction performances of COVID-19 waves in different regions and different periods, focusing on point estimates, while further interpreting the uncertainties in different rapidly changing dynamics, focusing on probabilistic predictions. The comparisons against different benchmarks could also provide additional insights on the importance of a particular exogenous data and model structures of the developed forecasting framework. More guidelines on COVID-19 forecast reporting and evaluation metrics can be found in [80, 81].”
Comment: Elaborate on the deep learning models.
Response: We thank the reviewer for the suggestion on elaborating the deep learning models section. We have now added additional high-level explanations on LSTM, which is the deep learning model incorporated by [45, 46]. We have also added details of the vanilla LSTM architecture in the Supplementary Materials section 2.1. The edited portion of section 4.1.2 Deep Learning Models is quoted as follows.
“LSTM models are a type of recurrent neural networks (RNN) [78], are able to overcome the drawbacks of the vanilla RNN, i.e., the problems of exploding and diminishing gradients, by additionally “remembering” portions of past, so called memory [76]. Therefore, LSTMs are capable of learning and capturing sophisticated relationships between the target and input data over long time lags in between, and produce robust and accurate forecasts. Further details of the vanilla LSTM structure are demonstrated in Supplementary Materials section 2”
Comment: Results should be better explained to show to the research community about the outcomes of the study.
Response: We thank the reviewer for pointing out the lack of clarity in the results section. We have now revised and reorganized the Results section (all changes are in color Blue). Specifically, we added additional descriptions on the general evaluation and results analysis procedures in the selected studies. Then, we divide the results into two subsections: 5.1 Importance of Internet Search Component and 5.2 Internet Search Information Serving as Early-Warning Signals.
Comment: Results should be more presentable for better understanding.
Response: We thank the reviewer for pointing out a better structure for the Results section. We have added additional structure in the Results section. Section 5.1 Importance of Internet Search Component now reads
“Although the forecasting targets, regions, time frames, and benchmark comparisons may differ from among all considered studies, they all demonstrate the importance of the internet search data component in their forecasting models through detailed sensi-tivity and evaluation analysis. In general, the sensitivity analysis is conducted by comparing the proposed models’ forecasts against other alternative benchmarks’ fore-casts, that have similar model structure but do not take internet search data into account. A few of the selected studies take a step further to compare against benchmarks with alternative model structure, to further illustrate internet search data’s additional ro-bustness and accuracy. … ”
Section 5.2 Internet Search Information Serving as Early-Warning Signals now reads
“Forecasting COVID-19 trends during its initial outbreak and subsequent waves is extremely challenging, due to noisy and unreliable groundtruth, limited historical data, and most notably the evolving nature of COVID-19 outbreaks. Meanwhile, internet search data could serve as early-warning detection signals for future outbreaks, and “foresee” surges of upcoming COVID-19 waves (Figure 1)...”
Reviewer 2 Report
This survey paper revolves around using Internet data to forecast the COVID-19 situation. The topic is interesting; however, some concerns must be addressed.
There are some sentences scattered all throughout the manuscript, which are a bit controversial and may be speculative.
The authors should revisit Table 1 and review the literature. It is not coherent, and the grouping seems rather opportunistic rather than well-planned or information-centric.
The readability, structure, and organization of the manuscript should be improved.
Some references that should have been cited are missing. The following papers are relevant and can be included.
· Mohammadhossein Ghahramani and Francesco Pilla, "Leveraging artificial intelligence to analyze the COVID-19 distribution pattern based on socio-economic determinants", Sustainable Cities and Society, 2021.
· J. Balest, A.E. Stawinoga, “Social practices and energy use at home during the first Italian lockdown due to Covid-19”, Sustainable Cities and Society, 2021.
Please add a discussion regarding the dimensionality of datasets used and how such data are being processed.
Some paragraphs do not read well. Please read through the paper and correct the typos.
I think the authors should include some detailed information regarding various proposed methods. The authors should present accurate evidence regarding all statements by providing detailed information relevant to the proposed solutions in the literature. The paper should provide a review of methods and techniques used in analyzing Internet data.
Future research perspectives should be enhanced from different angles.
Author Response
Summary of Changes in this Revision:
- We have re-organized the Results section, according to the suggestion from reviewer #1.
- We have an additional section 2.2 to briefly describe the search strategy and selection criteria of the studies considered in this review paper, according to the suggestion from reviewer #3. All details in the main manuscript are updated accordingly.
- We have added more detailed descriptions for deep learning models and statistical models in section 4, and Supplementary Materials section 2, according to the suggestions from reviewer #1 and reviewer #1.
- We have moved the evaluation procedures of forecasting models to an independent section, section 4.2, and added more descriptions on each error metrics, according to the suggestion from reviewer #1 and #3. All details in the main manuscript are updated accordingly.
- We have revised the section 5 Results, and re-organized the section for clarity, according to the suggestions from reviewer #1. All details in the main manuscript are updated accordingly.
- We have added COVID-19 targets’ sample dataset in the Supplementary Materials section 1, according to the suggestion from reviewer #1 and #2. All details in the main manuscript are updated accordingly.
- We have added additional descriptions for contributions of this review study and its viability in reapplying the methods in the Discussion section and Supplementary Materials section 3, according to the suggestion from reviewer #4.
- We have added additional future directions discussion in the Discussion section, according to the suggestions from reviewer #2.
Replies to Comments Provided by Reviewer #2
Summary: This survey paper revolves around using Internet data to forecast the COVID-19 situation. The topic is interesting; however, some concerns must be addressed.
Response: We thank the reviewer for all the suggestions. The detailed comments greatly improve the clarity of the manuscript. We have revised the paper accordingly, and all the changes are written in color blue.
Comment: There are some sentences scattered all throughout the manuscript, which are a bit controversial and may be speculative.
Response: We thank the reviewer for pointing out the potential controversy in the manuscript. We have carefully read through the paper and made changes accordingly. Below are a few places that we we edited:
- In section 1, the original sentence reads as: “Therefore, this study aims to present the extent of internet search data’s applications in a single document to enable future research studies to reciprocate and contribute to the field of COVID-19 forecasting and infectious diseases tracking more efficiently.” is changed to: “Therefore, this study aims to summarize the extended application of internet search data to enable future research studies to reciprocate and contribute to the field of COVID-19 forecasting and infectious diseases tracking more efficiently.”
- In section 3.2.2, the original sentence reads as “Google Trends’ internet search queries’ frequencies from Google Trends are sparse …” is changed to “The frequency of the obtained internet search queries from Google Trends are sparse.”
- In section 3.3, “There is a baseline value for everyday in a week, which represents the usual community mobility value…” is changed to “A baseline mobility index is established for 7 days in a week, …”
- In section 4.1.1, “..., and strong interpretability of the relationship between the time series to itself and to all exogenous variables incorporated.” changed to “..., and strong interpretability of autocorrelation and the relationship with all exogenous variables incorporated.”
Comment: The authors should revisit Table 1 and review the literature. It is not coherent, and the grouping seems rather opportunistic rather than well-planned or information-centric.
Response: We thank the reviewer for the lack of coherency in the selected review studies in this review paper. We have now added an additional section 2.2 to briefly describe the search strategy and selection criteria of the studies considered in this review paper, as well as re-emphasizing the goal of this review study. The added section 2.2 is also quoted below:
“In this section, the process to identify the articles for this review study is presented. Since this review study focuses on COVID-19 forecasting/prediction literature that utilizes internet search information, two constraints are imposed on the scope of this systematic and quantitative review study:
- Forecasting studies on population-level dynamics of COVID-19: papers that provide future predictions/forecasts for a specific region in the world. The search terms used are as follows: COVID-19, Coronavirus, SARS-COV-2, prediction models, forecasting models, predictive analysis.
- Data-driven including internet search: broadly defined as papers that incorporated COVID-19 related data, internet search data, and other exogenous information into the setup or fitting of the model. Here, the internet search information is broadly defined as datasets that reflect the online search behaviors of a population of interest. The search terms used are as follows: , internet search data, internet search information, Google Trends, online search behavior.
Various combinations of the search terms from each imposed were used to retrieve resources in particular databases, including Google Scholar [53], Scopus [54], and National Library of Medicine [55]. Some of the search strings used are as follows: “COVID-19 prediction models” AND “Internet search data”; “COVID-19 forecasting Models” AND “Google Trends”; “COVID-19 prediction” AND “Internet search information” AND “Mobility data”; “COVID-19forecast” AND “Online search behavior” AND “Time series information”. After the initial search, 217 documents were retrieved, of which 28 were duplicates. Then, by further filtering with both constraints above, 9 studies were finally selected for this review.
Table 1 lists out all the research studies considered in this paper, and their high-level overview of objectives, model types, data source, and quality assessments.”
Comment: The readability, structure, and organization of the manuscript should be improved.
Response: We thank the reviewer for pointing out the potential readability and organization issue in our manuscript. We have made the following changes:
- We have now revised and reorganized the Results section (all changes are in color Blue). Specifically, we added additional descriptions on the general evaluation and results analysis procedures in the selected studies. Then, we divide the results into two subsections: 5.1 Importance of Internet Search Component and 5.2 Internet Search Information Serving as Early-Warning Signals. A portion of section 5.1 is quoted as follows:
“Although the forecasting targets, regions, time frames, and benchmark comparisons may differ from among all considered studies, they all demonstrate the importance of the internet search data component in their forecasting models through detailed sensi-tivity and evaluation analysis. In general, the sensitivity analysis is conducted by comparing the proposed models’ forecasts against other alternative benchmarks’ fore-casts, that have similar model structure but do not take internet search data into account. A few of the selected studies take a step further to compare against benchmarks with alternative model structure, to further illustrate internet search data’s additional ro-bustness and accuracy. … ”
A portion of section 5.2 is quoted as follows:
“Forecasting COVID-19 trends during its initial outbreak and subsequent waves is extremely challenging, due to noisy and unreliable groundtruth, limited historical data, and most notably the evolving nature of COVID-19 outbreaks. Meanwhile, internet search data could serve as early-warning detection signals for future outbreaks, and “foresee” surges of upcoming COVID-19 waves (Figure 1)...”
- We have added an additional section 2.2 to briefly describe the search strategy and selection criteria of the studies considered in this review paper, enhancing the structure and readability of the paper. For details, please refer to the responses to the above comment.
- We have now moved the evaluation discussions to a separate section, section 4.2, and added probabilistic predictions’ error metrics, discussions on the evaluation process and references on COVID-19 forecasts’ evaluation guidelines. A portion of the revised section is also quoted below:
“To evaluate the accuracy of a probabilistic prediction and confidence or prediction interval of COVID-19 target (cases, deaths, and hospitalizations) against the actual groundtruth, the selected studies use two different evaluation metrics:the weighted interval score (WIS) and the empirical coverage. The weighted interval score (WIS) [79] is a proper scoring rule (smaller is better), that takes an entire predictive distribution into account and penalizes over- and under-confidence. Following CDC Forecast Hub's submission guideline [11] in this study, the WIS between the true value and a predictive distribution at time is evaluated across 11 prediction intervals with nominal coverages of 98%, 95%, 90%, …, 10%. The WIS between the true value and predictive distributions over the period t=1… T is computed by averaging the WIS across the period. The empirical coverage between the true value and the prediction intervals over period t=1…T is the proportion of true values falling inside a given central prediction interval with 95% nominal coverage.
The evaluation process is generally performed as follows: (1) the point estimates (or probabilistic predictions) of a particular COVID-19 target in a specific region and time interval of interest is produced from the developed model; (2) A particular evaluation error metric is chosen and the corresponding error is computed; (3) alternative benchmark point estimates (or probabilistic predictions) of the same COVID-19 target in the same region and time interval is collected; (4) benchmark error is computed with the same evaluation error metric; (5) performance analysis is conducted by comparing the error reduction between developed model and benchmark.
By choosing the appropriate benchmark, one can directly evaluate the prediction performances of COVID-19 waves in different regions and different periods, focusing on point estimates, while further interpreting the uncertainties in different rapidly changing dynamics, focusing on probabilistic predictions. The comparisons against different benchmarks could also provide additional insights on the importance of a particular exogenous data and model structures of the developed forecasting framework. More guidelines on COVID-19 forecast reporting and evaluation metrics can be found in [80, 81].”
- We have added more descriptions on Deep Learning models, and statistical models. Specifically, we added more high-level explanations on LSTM, which is the deep learning model incorporated by [40, 41]. We have also added details of the vanilla LSTM architecture in the Supplementary Materials section 2.1. The edited portion of section 4.1.2 Deep Learning Models is quoted as follows.
“LSTM models are a type of recurrent neural networks (RNN) [78], are able to overcome the drawbacks of the vanilla RNN, i.e., the problems of exploding and diminishing gradients, by additionally “remembering” portions of past, so called memory [76]. Therefore, LSTMs are capable of learning and capturing sophisticated relationships between the target and input data over long time lags in between, and produce robust and accurate forecasts. Further details of the vanilla LSTM structure are demonstrated in Supplementary Materials section 2”
Comment: Some references that should have been cited are missing. The following papers are relevant and can be included.
- Mohammadhossein Ghahramani and Francesco Pilla, "Leveraging artificial intelligence to analyze the COVID-19 distribution pattern based on socio-economic determinants", Sustainable Cities and Society, 2021.
- J. Balest, A.E. Stawinoga, “Social practices and energy use at home during the first Italian lockdown due to Covid-19”, Sustainable Cities and Society, 2021.
Response: We thank the reviewer for pointing out the potential lack of references. Both of the suggested references are cited in the edited paper now. The first reference above is cited among other newly included references in section 2.1, which now reads as:
“... Data-driven approach treats COVID-19 prediction as a time series prediction task, using available historical data and dynamic social behaviors. These models are typically built upon statistical frameworks [29-31] and recent advances in machine learning [32, 33] and deep learning algorithms [33-35].”
Second reference above is cited in section 1, which now reads:
“…, leading to drastic surges in confirmed cases, hospital admissions, and deaths, which severely threatened the health care systems and resources [2, 3].”
Comment: Please add a discussion regarding the dimensionality of datasets used and how such data are being processed.
Response: We thank the reviewer for the suggestion on expanding the discussion on internet search datasets. We have added more details in the Supplementary Materials section 1.2. We have also added table S3 in Supplementary Materials to showcase the sample dataset obtained from Google Trends. We have also revised the section 3.2.2 for more details of data preprocessing.
Comment: Some paragraphs do not read well. Please read through the paper and correct the typos.
Response: We thank the reviewer for the suggestion on improving readability of the paper. We have carefully read through the paper and corrected all mistakes to the best of our effort. We have also changed the organizations of multiple sections to improve readability. Please see the above responses to the other comments.
Comment: I think the authors should include some detailed information regarding various proposed methods. The authors should present accurate evidence regarding all statements by providing detailed information relevant to the proposed solutions in the literature. The paper should provide a review of methods and techniques used in analyzing Internet data.
Response: We thank the reviewer for the suggestion on expanding the model details of the selected studies, and the techniques in analyzing the internet search data. We have made the following edits according to the suggestions:
- We have added more descriptions on Deep Learning models in Supplementary materials section 2.1. Specifically, we added more high-level explanations on LSTM, which is the deep learning model incorporated by [40, 41]. We have also added details of the vanilla LSTM architecture in the Supplementary Materials section 2. The edited portion of section 4.1.2 Deep Learning Models is quoted as follows.
“LSTM models are a type of recurrent neural networks (RNN) [78], are able to overcome the drawbacks of the vanilla RNN, i.e., the problems of exploding and diminishing gradients, by additionally “remembering” portions of past, so called memory [76]. Therefore, LSTMs are capable of learning and capturing sophisticated relationships between the target and input data over long time lags in between, and produce robust and accurate forecasts. Further details of the vanilla LSTM structure are demonstrated in Supplementary Materials section 2”
- We have also added more descriptions on statistical models in Supplementary materials section 2.2, and explicitly expand on one selected statistical model in the selected review studies. The manuscript is also edited accordingly.
Comment: Future research perspectives should be enhanced from different angles.
Response: We thank the reviewer for the suggestion on future directions. We have added a paragraph at the end of the Discussion section regarding future directions.
“Many future directions remain open for internet search data applicability and infectious disease modeling, beyond addressing current limitations stated above. On the data modeling front, combining mechanistic models with data-driven statistical or deep learning models could be an interesting future direction. Examples include physics-informed neural networks with ordinary differential equation constraints [85], among others. On the disease dynamics front, as the prolonged COVID-19 pandemic continues, different infectious diseases may happen alongside the COVID-19 in different regions, including severe seasonal influenza [86] and monkeypox [87], causing additional burdens on health care resources and public safety. Accurate and real-time joint predictions of various infectious diseases in different geographical resolutions in a reliable and timely manner is an urgent and interesting future direction.”
Reviewer 3 Report
I enjoyed reading this review and have learned a lot about disease forecasting methods and about their relevance to health research. It was particularly interesting to see the wide spectrum of methods used in disease forecasting tasks. My comments to the authors are as follows:
1. I am not sure how you selected the 9 studies listed in Table 1. The study is titled as “a systematic review”. Identifying the relevant literature is crucial for a systematic review. Please clarify the process of identifying the studies included in your review (article databases and search terms).
2. Figure 1: This figure shows time-series, but the source of the data is not provided. Please add a reference to the original study. Or if this is from your own unpublished studies, then clarify the source.
3. Page 4, lines 328-329: Please help your readers and clarify “nonlinearities”.
4. Page 9, lines 358-383: You list several measures of predictive performance of the estimated models. Results are reported as error reductions. But do these metrics answer to the following question: How good are predictions? I think they do not allow to measure the goodness by a number we can compare across different applications. In linear regression models, reported R2 (the coefficient of determination). This measure R2 is more (intuitively) informative than MAE, MAPE, MSE, and RMSE in regression analysis evaluation, as the former can be expressed as a percentage, whereas the latter measures have arbitrary ranges. Consider discussing this drawback of the used metrics.
Author Response
Summary of Changes in this Revision:
- We have re-organized the Results section, according to the suggestion from reviewer #1.
- We have an additional section 2.2 to briefly describe the search strategy and selection criteria of the studies considered in this review paper, according to the suggestion from reviewer #3. All details in the main manuscript are updated accordingly.
- We have added more detailed descriptions for deep learning models and statistical models in section 4, and Supplementary Materials section 2, according to the suggestions from reviewer #1 and reviewer #1.
- We have moved the evaluation procedures of forecasting models to an independent section, section 4.2, and added more descriptions on each error metrics, according to the suggestion from reviewer #1 and #3. All details in the main manuscript are updated accordingly.
- We have revised the section 5 Results, and re-organized the section for clarity, according to the suggestions from reviewer #1. All details in the main manuscript are updated accordingly.
- We have added COVID-19 targets’ sample dataset in the Supplementary Materials section 1, according to the suggestion from reviewer #1 and #2. All details in the main manuscript are updated accordingly.
- We have added additional descriptions for contributions of this review study and its viability in reapplying the methods in the Discussion section and Supplementary Materials section 3, according to the suggestion from reviewer #4.
- We have added additional future directions discussion in the Discussion section, according to the suggestions from reviewer #2.
Replies to Comments Provided by Reviewer #3
Summary: I enjoyed reading this review and have learned a lot about disease forecasting methods and about their relevance to health research. It was particularly interesting to see the wide spectrum of methods used in disease forecasting tasks. My comments to the authors are as follows
Response: We thank the reviewer for the summary, the encouraging words, and for providing insightful and constructive comments, which greatly assist the draft’s quality and clarity. We have revised the paper accordingly (all changes are written in blue) and addressed all concerns and comments raised by the reviewer via point-to-point responses below.
Comment: I am not sure how you selected the 9 studies listed in Table 1. The study is titled “a systematic review”. Identifying the relevant literature is crucial for a systematic review. Please clarify the process of identifying the studies included in your review (article databases and search terms).
Response: We thank the reviewer for pointing out the selection process of the literatures included in our review. Indeed, it is important to include the identification process of these included studies. We have now added an additional section 2.2 to briefly describe the search strategy and selection criteria of the studies considered in this review paper. The added section 2.2 is also quoted below:
“In this section, the process to identify the articles for this review study is presented. Since this review study focuses on COVID-19 forecasting/prediction literature that utilizes internet search information, two constraints are imposed on the scope of this systematic and quantitative review study:
- Forecasting studies on population-level dynamics of COVID-19: papers that provide future predictions/forecasts for a specific region in the world. The search terms used are as follows: COVID-19, Coronavirus, SARS-COV-2, prediction models, forecasting models, predictive analysis.
- Data-driven including internet search: broadly defined as papers that incorporated COVID-19 related data, internet search data, and other exogenous information into the setup or fitting of the model. Here, the internet search information is broadly defined as datasets that reflect the online search behaviors of a population of interest. The search terms used are as follows: , internet search data, internet search information, Google Trends, online search behavior.
Various combinations of the search terms from each imposed were used to retrieve resources in particular databases, including Google Scholar [53], Scopus [54], and National Library of Medicine [55]. Some of the search strings used are as follows: “COVID-19 prediction models” AND “Internet search data”; “COVID-19 forecasting Models” AND “Google Trends”; “COVID-19 prediction” AND “Internet search information” AND “Mobility data”; “COVID-19forecast” AND “Online search behavior” AND “Time series information”. After the initial search, 217 documents were retrieved, of which 28 were duplicates. Then, by further filtering with both constraints above, 9 studies were finally selected for this review.
Table 1 lists out all the research studies considered in this paper, and their high-level overview of objectives, model types, data source, and quality assessments.”
Comment: Figure 1: This figure shows time-series, but the source of the data is not provided. Please add a reference to the original study. Or if this is from your own unpublished studies, then clarify the source.
Response: We thank the reviewer for pointing out the lack of references for figure 1. We have added the references for all the raw data plotted in the figure 1 caption. We want to mention that all the plotted lines are directly downloaded from the cited references included.
We have also included the additional plot of delay in peak between the search frequency of “loss of taste” to COVID-19 hospitalizations, as some of the included studies in our review paper also have hospitalization as their forecast target.
Comment: Page 9, lines 328-329: Please help your readers and clarify “nonlinearities”.
Response: We thank the reviewer for pointing out the lack of explanation for “nonlinearities”. We have now added an additional sentence to briefly explain the “nonlinearities” in the disease forecasting concept, as well as related references. The modified portion of the paragraph is also quoted below:
“An example of nonlinearity in COVID-19 forecasts is that the number of cases lead to a disproportionate number of deaths during the peaking periods of a COVID-19 wave [2]. More technical details on nonlinearities in time series are discussed in [72].”
Comment: Page 9, lines 358-383: You list several measures of predictive performance of the estimated models. Results are reported as error reductions. But do these metrics answer the following question: How good are predictions? I think they do not allow us to measure goodness by a number we can compare across different applications. In linear regression models, reported FFR2 (the coefficient of determination). This measure R2 is more (intuitively) informative than MAE, MAPE, MSE, and RMSE in regression analysis evaluation, as the former can be expressed as a percentage, whereas the latter measures have arbitrary ranges. Consider discussing this drawback of the used metrics.
Response: We thank the reviewer for pointing out the lack of explanation and analysis of the evaluation metrics. We would like to point out that the point estimate error metrics (RMSE, MAE, MAPE, and correlation), and probabilistic prediction error metrics (WIS, and coverage), that we discussed in the review paper, are fundamentally different than R2 (the coefficient of determination). The former metrics are used to evaluate the forecasts/predictions on the unseen data that are not a part of fitting/training the model, while the latter measurement is used to evaluate the fitting and goodness of the model and its parameters. In short, a high R2 of a particular forecasting model might not lead to low forecast errors, but only infer that the model correctly explained the target’s relationship with the exogenous variables partially. In order to fully evaluate the forecasting models’ performances in different periods and regions, one ought to train the model up to (or prior to) the time period of interest, forecast the COVID-19 target during the time period of interest, and evaluate the forecasts with the actual groundtruth using one or multiple error metrics above.
We have now moved the evaluation paragraphs to a new section, section 4.2, and additionally included the evaluation procedure and references on forecasting evaluation guidelines. The paper is revised accordingly, and the revised portion is also quoted below:
“The evaluation process is generally performed as follows: (1) the point estimates (or probabilistic predictions) of a particular COVID-19 target in a specific region and time interval of interest is produced from the developed model; (2) A particular evaluation error metric is chosen and the corresponding error is computed; (3) alternative benchmark point estimates (or probabilistic predictions) of the same COVID-19 target in the same region and time interval is collected; (4) benchmark error is computed with the same evaluation error metric; (5) performance analysis is conducted by comparing the error reduction between developed model and benchmark.
By choosing the appropriate benchmark, one can directly evaluate the prediction performances of COVID-19 waves in different regions and different periods, focusing on point estimates, while further interpreting the uncertainties in different rapidly changing dynamics, focusing on probabilistic predictions. The comparisons against different benchmarks could also provide additional insights on the importance of a particular exogenous data and model structures of the developed forecasting framework. More guidelines on COVID-19 forecast reporting and evaluation metrics can be found in [80,81].”
Reviewer 4 Report
This a very interesting approach in showcasing public search behavior as a valuable alternative data source assisting short-term forecasts when the amount of data is limited like during the pandemic. The recommendation is to extend on describing the contributions of your study and its viability in reapplying your method and whether other researchers will get the same results.
Author Response
Summary of Changes in this Revision:
- We have re-organized the Results section, according to the suggestion from reviewer #1.
- We have an additional section 2.2 to briefly describe the search strategy and selection criteria of the studies considered in this review paper, according to the suggestion from reviewer #3. All details in the main manuscript are updated accordingly.
- We have added more detailed descriptions for deep learning models and statistical models in section 4, and Supplementary Materials section 2, according to the suggestions from reviewer #1 and reviewer #1.
- We have moved the evaluation procedures of forecasting models to an independent section, section 4.2, and added more descriptions on each error metrics, according to the suggestion from reviewer #1 and #3. All details in the main manuscript are updated accordingly.
- We have revised the section 5 Results, and re-organized the section for clarity, according to the suggestions from reviewer #1. All details in the main manuscript are updated accordingly.
- We have added COVID-19 targets’ sample dataset in the Supplementary Materials section 1, according to the suggestion from reviewer #1 and #2. All details in the main manuscript are updated accordingly.
- We have added additional descriptions for contributions of this review study and its viability in reapplying the methods in the Discussion section and Supplementary Materials section 3, according to the suggestion from reviewer #4.
- We have added additional future directions discussion in the Discussion section, according to the suggestions from reviewer #2.
Replies to Comments Provided by Reviewer #4
Summary: This a very interesting approach in showcasing public search behavior as a valuable alternative data source assisting short-term forecasts when the amount of data is limited like during the pandemic.
Response: We thank the reviewer for the insightful comments and encouraging words. We have revised the paper accordingly (all changes are written in blue), and addressed the comments raised by the reviewer via point-to-point responses below. The constructive suggestions have greatly helped us improve the clarity and quality of our draft and are highly appreciated.
Comment: The recommendation is to extend on describing the contributions of your study and its viability in reapplying your method and whether other researchers will get the same results.
Response: We thank the reviewer for the recommendation to extend on the contributions of our studies. We have made the following modifications to our manuscript:
- We have added table S3 in Supplementary Materials section 3, that organizes the data and code availability of the 9 selected research studies. The majority of the studies deposited their data and code used to produce study results to publicly available URLs for access. This enhances the applicability and reproducibility of the research studies. We have also added additional sentences in section 3 Data Acquisition and Pre-processing, and section 4 Methods, referring to table S3 in Supplementary Materials for data and code availability. They are also quoted below:
“Details of data usage and availability of the selected research studies (Table 1) are further provided in Table S3 in Supplementary Materials.”
“This section briefly introduces the various prediction models used by the 9 identified studies (Table 1), as well as the subsequent evaluation process for the forecasting results. Details of model implementations and code availabilities of the selected studies are further provided in Table S3 in Supplementary Materials.”
- We have added additional sentences in the Discussion section to emphasize the general applicability and predictive power of internet search data for future diseases. The edited portion is also quoted as follows.
“Furthermore, as internet search information have been shown to track successfully with various diseases such as influenza [12, 16, 18, 21], dengue [17], and Zika [82,83], among others [84], the selected research studies in this paper provides additional applicability and predictive insights for the internet search data, and take a step further for future infectious diseases.”
Round 2
Reviewer 2 Report
The paper has been improved and the raised concerns have been addressed.